# CMOS-based bio-image sensor spatially resolves neural activity-dependent proton dynamics in the living brain

Hiroshi Horiuchi [1,2,4], Masakazu Agetsuma [1,4], Junko Ishida [1], Yusuke Nakamura[3], Dennis Lawrence Cheung [1], Shin Nanasaki[3], Yasuyuki Kimura [3], Tatsuya Iwata [3], Kazuhiro Takahashi[3], Kazuaki Sawada[3]* & Junichi Nabekura[1,2]*

Recent studies have shown that protons can function as neurotransmitters in cultured neurons. To further investigate regional and neural activity-dependent proton dynamics in the brain, the development of a device with both wide-area detectability and high spatial-temporal resolution is necessary. Therefore, we develop an image sensor with a high spatial-temporal resolution specifically designed for measuring protons in vivo. Here, we demonstrate that spatially deferent neural stimulation by visual stimulation induced distinct patterns of proton changes in the visual cortex. This result indicates that our biosensor can detect micrometer and millisecond scale changes of protons across a wide area. Our study demonstrates that a CMOS-based proton image sensor with high spatial and temporal precision can be used to detect pH changes associated with biological events. We believe that our sensor may have broad applicability in future biological studies.

[1] Division of Homeostatic Development, National Institute for Physiological Sciences, National Institutes of Natural Sciences, Okazaki 444-8585, Japan. [2] Department of Physiological Sciences, SOKENDAI: The Graduate University for Advanced Studies, Hayama 240-0193, Japan. [3] Department of Electrical and Electronic Engineering, Toyohashi University of Technology, Toyohashi 441-8580, Japan. [4] These authors contributed equally: Hiroshi Horiuchi, Masakazu Agetsuma. *email: sawada@ee.tut.ac.jp; nabekura@nips.ac.jp

The regulation of proton concentration (pH) in the brain is important for maintaining normal brain function. In the brains of healthy subjects, intracellular pH is maintained at 6.8–7.0, whereas extracellular pH is maintained at 7.2–7.4 (ref. [1]). The brain consists of neurons and glial cells, i.e. astrocytes, microglia, and oligodendrocytes, which all have various transporter proteins, such as the $Na^+/H^+$ exchanger, $Na^+$-driven $Cl^-/HCO_3^-$ exchanger, $Na^+/HCO_3^-$ cotransporter, and the passive $Cl^-/HCO_3^-$ exchanger, that collectively act to maintain intracellular and extracellular pH levels[1]. While the homeostatic importance of pH regulation has long been appreciated, more recent studies have shown that protons can also directly participate in neurotransmission[2]. This suggests an added dimension in terms of the relevance of pH changes to brain function under both physiological and pathological conditions[3].

However, further technical advances in pH measurement methods are required to create better probes with high spatial and temporal resolution to evaluate pH changes at neural circuit level. Double barreled and concentric microelectrodes can only measure pH at a single point, thus their utility is limited to correlating proton changes with globalized brain activity, for example, during seizures and ischemia[4,5]. In contrast, magnetic resonance imaging (MRI) is able to simultaneously measure the distribution of protons in the entire brain and is thus able to detect regional variations in pH. For example, presentation of a flashing checkerboard visual stimulus to human subjects induces pH changes localized to the visual cortex[6]. However, the spatial and temporal resolutions of MRI are limited to ~4 mm and ~6 s[6], respectively, which are too broad to study pH changes relevant to neurotransmission as these changes occur at spatial and temporal scales of micrometers and milliseconds.

To overcome these limitations, we develop a special proton image sensor device that is based on our previous $128 \times 128$ pixel CMOS-based proton image sensor[7], but specifically optimized for in vivo brain analyses. We redesign the proton image sensor to have a slimmer chip width and reduced thickness in such a way that inserting it into the brains of live animals only causes very minimal damage to the surrounding brain tissue. We demonstrate that our proton image sensor can make sensitive and accurate pH measurements at a high spatial–temporal resolution and subsequently use it to measure localized pH changes in the brains of live mice, specifically in the primary visual cortex (V1) area, while they undergo a visual experience task. Because we are able to measure pH changes at micrometer and millisecond scales of resolution, we are able to correlate distinct spatial patterns of pH changes in the V1 with different visual stimulus patterns, suggesting that our device may be useful in gaining a deeper insight into the relationship between pH changes and computation in neural circuits.

## Results

### Development of an insertion-type proton image sensor.
In the present study, we redesigned the $128 \times 128$ pixel CMOS-based proton image sensor[7] to be suitable for in vivo experiments (Fig. 1). Thus, the sensor chip size was decreased to a final dimension of 11.47 mm length × 1.76 mm width × 0.1 mm thickness, with $128 \times 32$ pixels in its pH-sensing area (Fig. 1b). The spatial and temporal resolutions of our proton image sensor were 23.55 μm × 23.55 μm per pixel and 50 frames $s^{-1}$, respectively, similar to the previous proton image sensor[7] (Fig. 1c). The characteristics of the proton image sensor are summarized in Table 1.

As in the previous proton image sensor, this sensor detects changes in proton concentration through the chemical equilibrium involving protons at the $Si_3N_4$ surface which alters the surface potential at each pixel[7]. Thus, its proton sensitivity is reflected in the size of the voltage increment per change in pH. For the five central pixels, the pH sensitivity was determined to be 51.6 mV $pH^{-1}$ (which includes readout circuit and charge transfer amplification) and thus surpassed that of the previous pH sensor (32.8 mV $pH^{-1}$) (Supplementary Fig. 1a, b)[7]. Furthermore, the pH voltage$^{-1}$ values of all pixels throughout the entire pH-sensing area were nearly identical, indicating low inter-pixel variation (Supplementary Fig. 1c, d). In a sensor, groups of pixel-by-pixel values at the three different pH conditions (pH 9.18, 6.81, and 4.01) did not overlap each other and were statistically distinct (Supplementary Fig. 1c). We further calculated the standard deviation (SD) over multiple devices ($n = 12$ sensors in total; SD = 0.0061 ± 0.0013 (pH 9.18), 0.0037 ± 0.0027 (pH 6.81), 0.0171 ± 0.0117 (pH 4.01) as illustrated in Supplementary Fig. 1d). These inter-pixel variations were relatively small when compared to the value (SD = 0.027) shown in the previous report[8]. These results suggest that our sensor has low inter-pixel variation. These results suggest that the modifications incorporated into our present proton image sensor had no adverse impact on functionality, compared to our previous pH sensor[7].

### Visualized pH changes in live animals by visual stimulus.
Magnotta et al.[6] have previously demonstrated using MRI that visual stimuli trigger pH changes associated with brain activity in the visual cortex[6]. Thus, we used our proton image sensor to measure pH changes in the V1 in live mice while they underwent a visual experience task. Compared with MRI, our proton image sensor is able to detect such pH changes at a much higher spatial–temporal resolution[7].

After inserting the proton image sensor into the brain, we measured the depth of insertion to be approximately 2.0 mm from the brain surface using the photoelectric effect (see Section Insertion of the pH sensor into the visual cortex) (Supplementary Fig. 2). This indicated that the proton image sensor was correctly situated within the V1 which was subsequently confirmed by histology performed after the pH measurement experiments (Supplementary Fig. 3).

We evoked brain activity, and thus pH changes, in the V1 of mice by visual stimulation. We presented visual stimuli consisting of drifting gratings at various directions, for example, black and white bars sweeping across a screen at various defined angles (described in detail in section Visual stimulation) (Fig. 2a, b). Previous electrophysiological experiments showed that neurons in the V1 exhibited direction-selective responses, for example, the activity of individual neurons strongly increases in response to particular directions of stimulus movement[9–11]. Importantly, the direction selectivity of neurons is highly heterogeneous and intertwined in the mouse V1 area[12]. Thus, we measured responses in the V1 during drifting gratings of multiple angles as an appropriate challenge for testing the spatial–temporal resolution capabilities of our proton image sensor. As expected, our proton image sensor with its resolution of micrometers and milliseconds captured distinct spatial patterns of pH changes in the V1 induced by each of the eight differently oriented drifting gratings (Fig. 2c). Similar experiments were performed in a HEPES-buffered saline solution (pH 7.4) instead of in live mice to confirm that these pH changes were not just artifacts but reflected actual changes in brain pH. As expected, there were no significant changes in pH observed in the HEPES-buffered saline experiments (Fig. 2d). These results indicate that our proton image sensor can successfully detect pH changes relevant to neurotransmission, as induced by visual stimuli, in a localized brain area with a high spatial resolution.

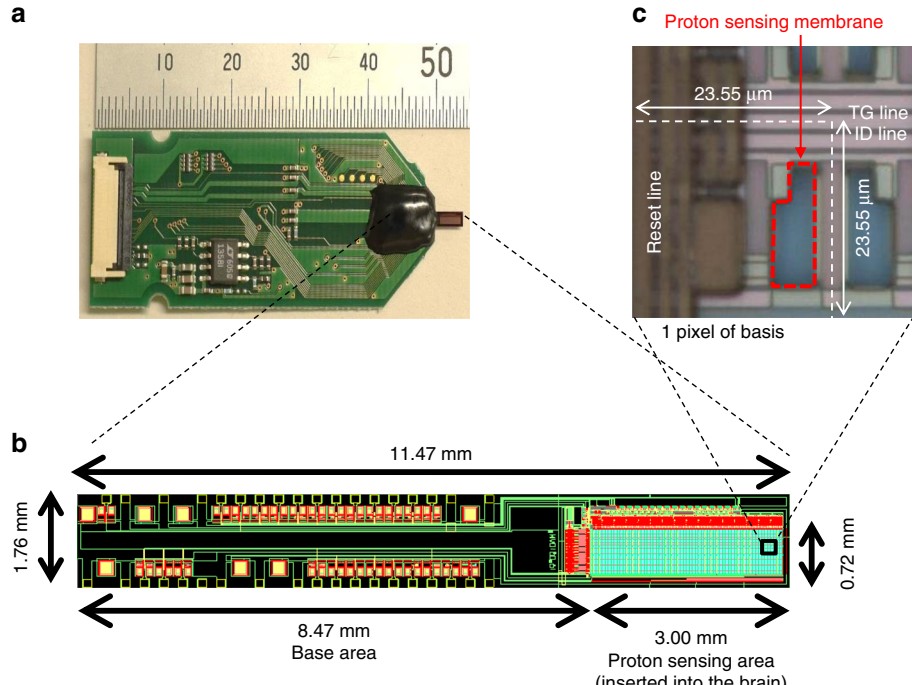

**Fig. 1 Structure of the pH sensor. a** Photograph of the complete proton image sensor which consists of the sensor chip packaged with its printed circuit board (PCB). The sensor chip detects changes in pH and is connected to the PCB by wire bonds covered in black epoxy. **b** Sensor chip layout which defines two areas, the proton-sensing area and the base area. The proton-sensing area is responsible for detecting changes in pH and is inserted into the mouse brain while the base area contains the connections between the proton-sensing area and the PCB. **c** Micrograph of the pH sensor pixels. Each pixel is able to independently detect changes in pH. Thus, the proton-sensing area, which contains 128 × 32 pixels, is able to map fine-scale regional pH variations in the local environment.

| Table 1 Characterization of the proton image sensor. | | | |
| --- | --- | --- | --- |
| **Number of pixels** | **Size of pixel [μm²]** | **Size of sensor chip [mm³]** | **Frame rate [frames s⁻¹]** |
| 32 × 128 | 23.55 × 23.55 | 1.76 × 11.46 × 0.1 | 50 |

**Distribution of angle-specific pH changes by visual stimulus.** To more rigorously confirm whether the observed localized pH changes in the V1 reflected actual pH changes in the brain, we examined the distributions in the size of the pH changes detected at each pixel for the various drifting gratings directions (Fig. 3, representative examples). In live mice (Fig. 3a), a greater number of pixels showed large pH changes, which were dependent on the direction of the drifting gratings as compared with HEPES-buffered saline experiments (Fig. 3b). To quantify this, we counted the number of pixels with pH changes which exceeded threshold values (inside of red lines in Fig. 3) based on the 95% confidence interval for the mean change in pH in the HEPES-buffered saline experiments (summarized data of $n = 3$ experiments). This suggested that the direction in which pH was shifted depended on the particular direction of the drifting gratings pattern. In live mice, the distributions in the sizes of pH changes at individual pixels were shifted towards alkalinity in response to drifting gratings directions of 0°, 90°, and 270°, whereas these distributions were shifted towards acidity in response to drifting gratings directions of 45°, 135°, and 180° (Fig. 3a). In contrast, these direction-dependent pH changes were not observed in HEPES-buffered saline pH measurements, which was expected (Fig. 3b). Next, we pooled this data across all experiments ($n = 9$ experiments in brains, $n = 3$ in HEPES control), for each of the

drifting gratings directions (Fig. 4). This confirmed that our sensor was able to detect brain state-specific patterns of pH change that were induced by different drifting gratings directions.

We also used a different statistical method to determine whether the pH change recorded at a given pixel was a signal artifact or reflected an actual change in brain pH. Here, we quantified the number of pixels that showed statistically significant differences in their pH values during stimulus (drifting gratings) and during stimulus interval (gray screen). Pixels with such changes are indicated in Fig. 5a which is a representative example of the pH measurements in live mice. We then pooled these pixels across all animals for each of the different drifting gratings directions and categorized them based on whether they were inserted inside the brain or located outside the brain surface during the visual experience task (Fig. 5b). Two-way ANOVA analyses revealed a significant difference between the number of pixels within the brain and those located outside the brain surface (interaction: $F(1, 95) = 8.32$, $p = 0.005$). Furthermore, pooling across all drifting gratings angles also showed a statistically significant difference between the number of pixels within the brain and the number of pixels located outside the brain surface (Fig. 5c). Altogether these results suggest that the pH changes detected by our proton image sensor are indeed induced by the visual experience task and not the result of the signal artifact.

**Temporal change in pH by visual stimulation.** To investigate temporal change in pH triggered by visual stimulation more accurately, we summarized the data in peri-stimulus time histograms to visualize the dynamics before and after the onset of the visual stimulation (Supplementary Fig. 4). This is also a good measure to evaluate the temporal resolution of our sensor. We calculated the mean pH dynamics of each pixel before, during,

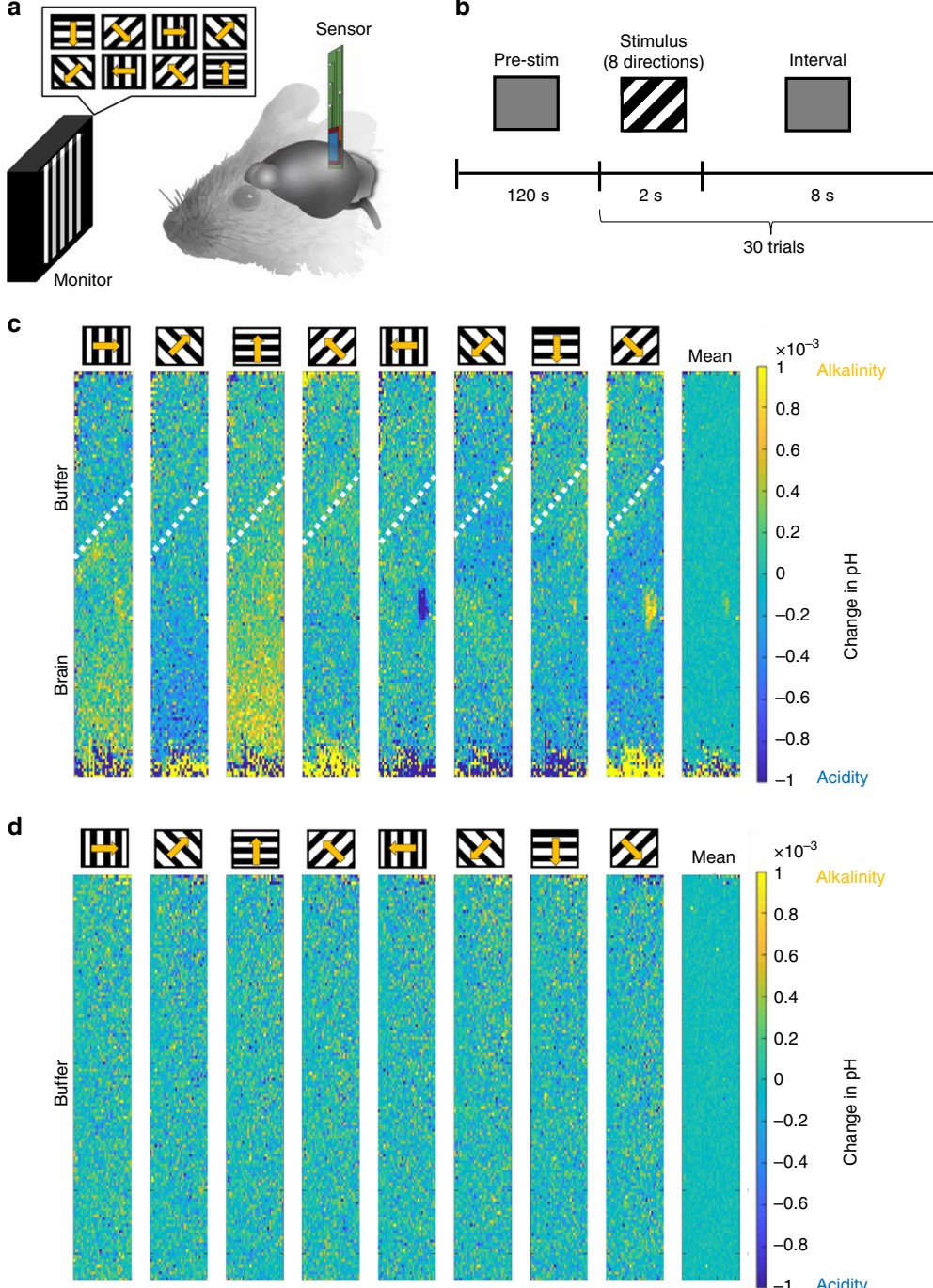

**Fig. 2 Detection of pH changes in the brain during visual stimulation. a** Visual stimuli consisted of drifting gratings with eight different directions, which were composed of white and black bars sweeping across a screen at eight different direction angles. The screen was positioned 13 cm away from the right eye of mice. Changes in pH were recorded from the primary visual cortex (V1) for each angle. **b** After a 120 s pre-stimulation phase (gray screen), the stimulation phase comprised a 2 s visual stimulus (one of eight drifting gratings directions was randomly selected each time) and an 8 s interval (gray screen) presented to the mice. This was repeated 30 times (trials) for each of the eight drifting gratings direction. **c, d** Observation of changes in pH caused by visual stimuli. Imaging at the pixel-by-pixel resolution revealed that different types of responses were induced by different directions of drifting gratings. The value at each pixel was calculated as the difference between the pH measured for each drifting gratings direction and the mean pH measured across all eight directions, in a representative mouse brain (**c**) or in representative HEPES-buffered saline (**d**). White dotted lines indicate borderlines between the brain and HEPES buffer, which were detected by photoelectric effect (see Supplementary Fig. 2). The source data underlying Fig. 2c, d are provided as a Source Data file.

and after the visual stimulation, over multiple trials (in Supplementary Fig. 4a, pre-stim (2 s interval before each stimulation), stim (2 s stimulation), and post-stim (4 s interval after each stimulation) respectively). The response pattern at each pixel was categorized as an alkaline response, acidic response, or neutral response based on whether the pH was statistically increased, decreased, or unchanged by visual stimulation. This clearly indicated the temporally dynamic change in pH at each pixel. To

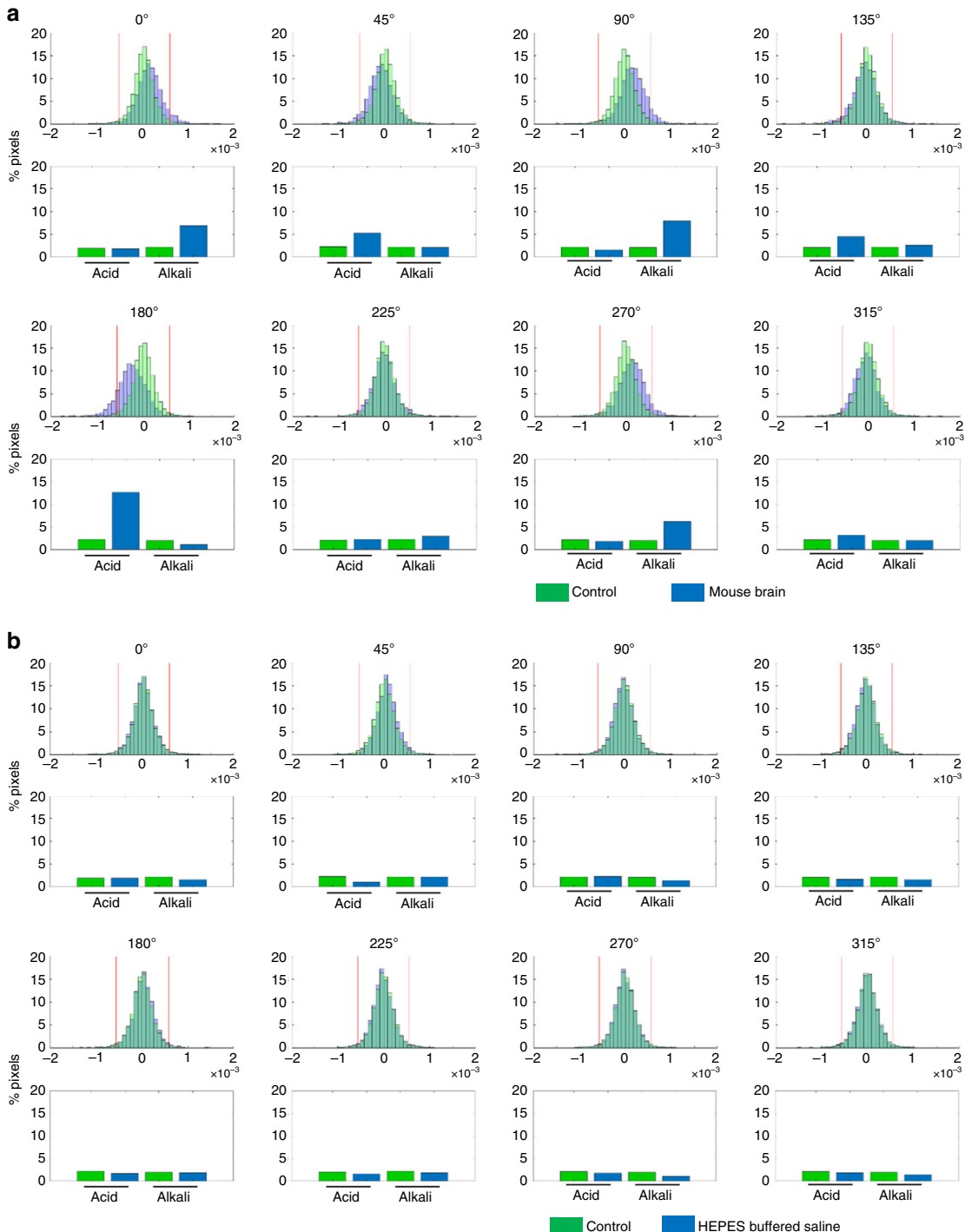

**Fig. 3 Stimulus type-dependent modification of pH states in the brain. a, b** Representative distribution histograms of the sizes of pH changes at individual pixels (referred to in the text as delta pH), induced by the eight different drifting gratings directions, recorded in a mouse brain (**a**, or in HEPES-buffered saline (**b**)). The green histograms represent the average delta pH values measured in all HEPES-buffered saline experiments (3 experiments, 8 × 30 visual stimulation trials per experiment). The blue histograms are representative examples of delta pH values taken from one mouse (8 × 30 visual stimulation trials), or one HEPES-buffered saline sample (8 × 30 visual stimulation trials). Note that the mouse brain delta pH distributions show either acidic or alkaline shifts (**a**) whereas the HEPES-buffered saline delta pH distributions do not (**b**). The red lines on each of the histogram plots represent the 95% confidence interval values calculated from the average delta pH values measured in all HEPES-buffered saline experiments. Using these as threshold values was one method of defining which pixels measured pH changes that reflected actual changes in brain pH.

quantify these responses in the brain, we summarized all response patterns from all angles and animals for each response category (Supplementary Fig. 4b, $n = 9$), and calculated the time constant of acidic and alkaline change in brain pH during visual stimulation

($\tau_{alkali, \; fast} = 250.2$ ms, $\tau_{alkali, \; slow} = 14.19$ s, $\tau_{acid, \; fast} = 231.0$ ms, $\tau_{acid, \; slow} = 6.99$ s). These results demonstrate that our sensor successfully detected the sub-second order temporal change in brain pH during visual stimulation.

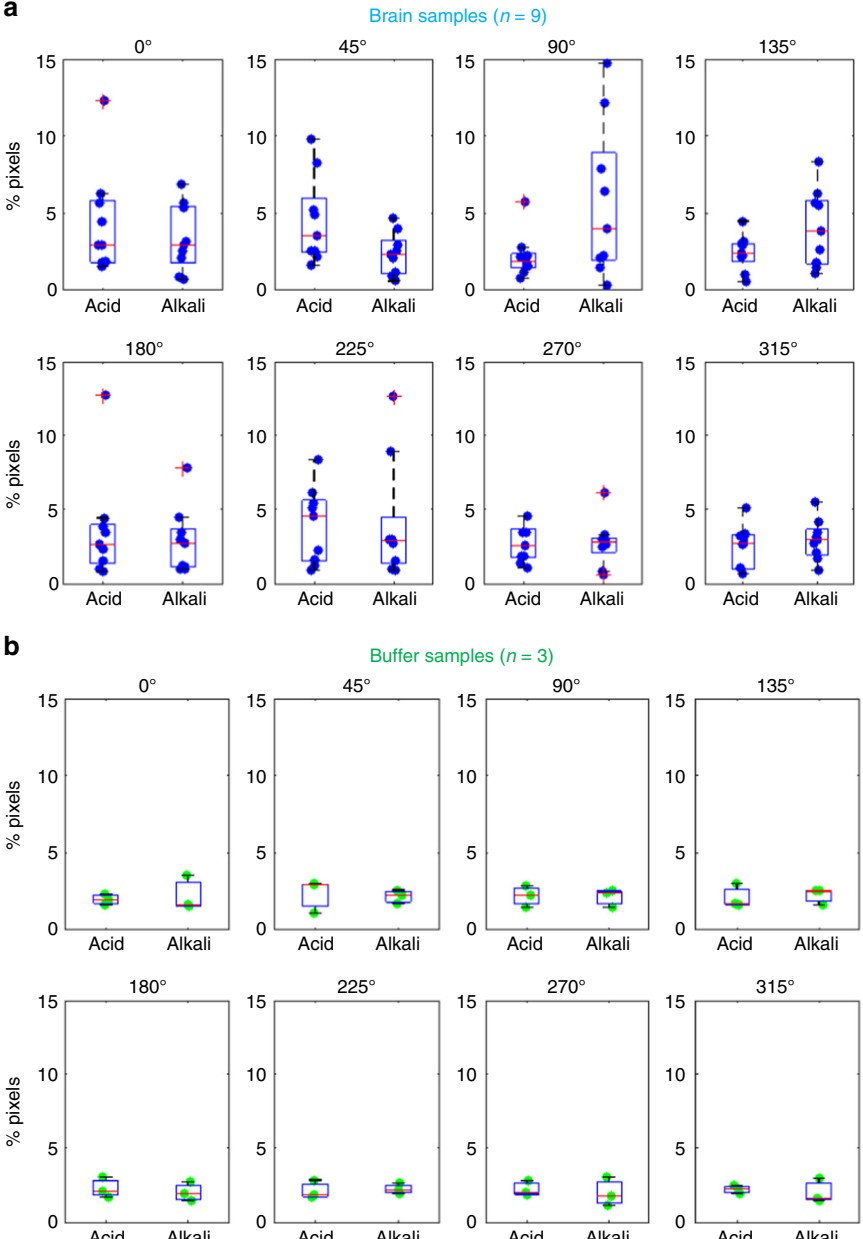

**Fig. 4 Significant pH changes were detected in the brain. a, b** The proportions of pixels with pH changes defined as actual changes in brain pH were pooled across all nine experiments performed in mouse brains (**a**) or all three experiments performed in HEPES-buffered saline (**b**). As described in Fig. 3, a pixel with a pH change that exceeded the 95% confidence interval values based on the mean delta pH value of all HEPES-buffered saline experiments (*n* = 3) was defined as showing an actual change in brain pH. Thus, the proportions of such pixels in the HEPES-buffered saline experiments (**b**) suggest the false-positive rate of this definition. In the box and whisker plots, the center line indicates the median; the box defines the 25–75th percentiles; the whiskers represent minimum and maximum values while the red crosshairs represent outliers. Filled circles are individual data points. Green plots correspond to HEPES-buffered saline while blue plots correspond to mouse brains. Source data are provided as a Source Data file.

## Discussion

In this study, we developed a pH sensor optimized for the observation of pH changes in the brains of live mice. To minimize tissue damage, we miniaturized the device down to 0.1 mm thickness, while, to detect the change in pH in a wide brain area, we kept a 128 × 32 pixel sensing area. This resulted in a 1.76 mm width of our device, which is still smaller than the previous 128 × 128 pixel sensor. Although it is larger in comparison to some of the electrodes used for neural activity or pH recording, the application of these devices is limited to the single point recording. Instruments for wide-field observation or recording with high spatial resolution tends to be larger[13–18]. For example,

the GRIN lens, which has been broadly used to investigate neural function in the deep and wide area of the brain (such as the amygdala, hippocampus, etc.), is usually of 0.6–1.0 mm diameter and inserted into the 2.0–5.0 mm or deeper areas[13,14]. Our device (0.176 mm² in cross-section) is theoretically less invasive than the regular GRIN lens (0.283–0.785 mm²).

We confirmed the reliability of this approach by observing the dynamic pH changes in the V1 that were induced by a visual experience task and which have been similarly reported in a previous MRI study[6]. Furthermore, given that our sensor has a much higher spatial and temporal resolution than MRI[6], we were able to successfully resolve the stimulus-dependent differences of

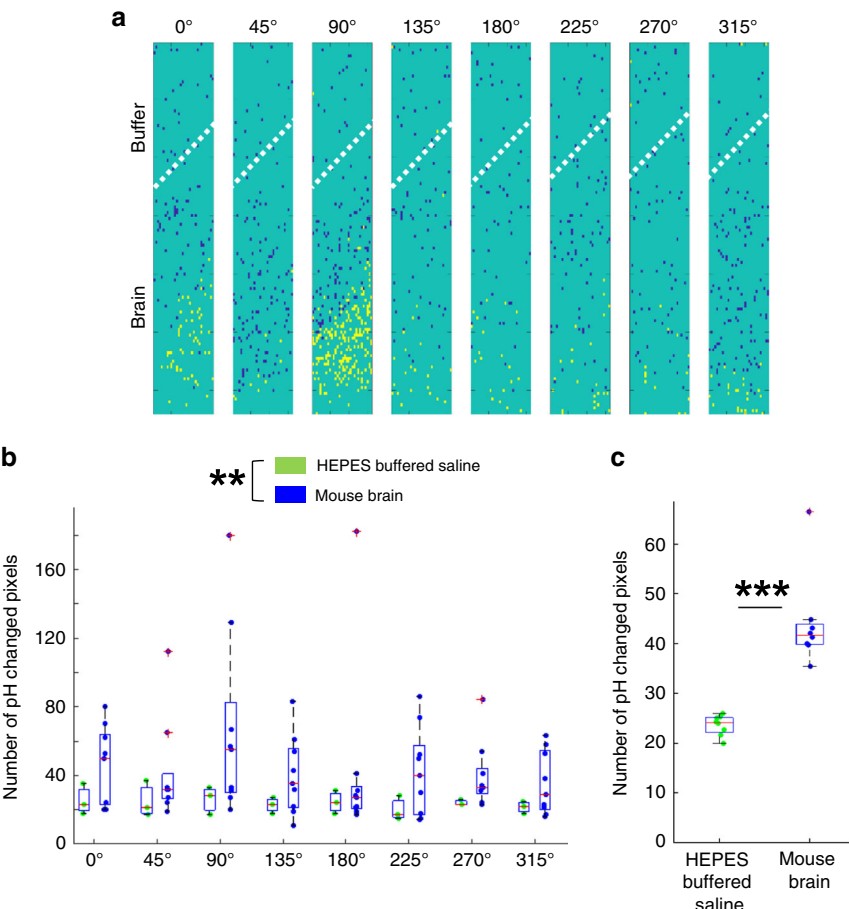

**Fig. 5 Pixel-by-pixel analysis proved significant pH change in the brain during visual stimulation. a** A representative example of the spatial distribution of pixels with statistically significant changes in pH during visual stimuli. The data in Fig. 2c were analyzed here. A one-sample *t*-test was used at each pixel to statistically compare the difference between mean pH change for each direction of drifting gratings (a total of 30 trials for each direction) and mean pH change for all eight directions (i.e. the mean over 8 directions × 30 trials). Pixels with a significant decrease (acidic change, $p < 0.05$) are shown in blue, while pixels with a significant increase (alkaline change, $p < 0.05$) are marked in yellow. The spatial distribution differed depending on the direction of the stimulation. White dotted bars demarcate the region of the pH-sensing area inserted into the brain and the region of the pH-sensing area left outside of the brain (see Supplementary Fig. 2). **b** The number of pixels with statistically significant changes in brain pH was summarized for all observations in mice ($n = 9$) or in HEPES-buffered saline ($n = 3$) for each direction of visual stimulation. A two-way ANOVA test indicated a statistically significant difference between mouse brain and HEPES-buffered saline measurements (**$p < 0.01$). In the box and whisker plots, the center line indicates the median; the box defines the 25–75th percentiles; the whiskers represent minimum and maximum values while the red crosshairs represent outliers. Filled circles are individual data points. Green plots correspond to HEPES-buffered saline while blue plots correspond to mouse brains. **c** The number of pixels of significant changes in pH was pooled across all eight drifting gratings directions. An unpaired *t*-test indicated a significant difference between brain and HEPES-buffered samples (***$p < 0.001$). Definitions of box and whisker plots and *n* numbers are same as in that stated in **b**. The source data underlying **b** and **c** are provided as a Source Data file.

pH changes in the brain (Figs. 2–5, Supplementary Fig. 4). Although the spatial coverage of our sensor is smaller than that of MRI (MRI: whole brain ($220 \times 220$ mm $= 4.84 \times 10^4$ mm$^2$), our sensor: 0.72 mm × 3.0 mm $= 2.16$ mm$^2$)[6], our sensor is advantageous for the spatial and temporal resolution (MRI: ~4.0 mm, ~0.17 frames s$^{-1}$, our sensor: 23.55 μm, 50 frames s$^{-1}$)[6]. Also, the spatial coverage of our sensor is larger than regular two-photon microscopy (2PMS), an imaging technique for deep brain recording, or single-photon imaging through a GRIN lens (our sensor: 2.16 mm$^2$, 2PMS: ~0.25 mm$^2$, GRIN lens: ~0.785 mm$^2$ with more invasion)[13,14,19]. Our sensor is also superior to the other imaging techniques in terms of temporal resolution (our sensor: 50 frames s$^{-1}$, 2PMS: 30 frames s$^{-1}$, MRI: ~0.17 frames s$^{-1}$, $^{31}$P spectrometry: several minutes per frame)[6,19,20]. Based on the advanced temporal resolution, we successfully demonstrated the detection of the sub-second order dynamics of the change in brain pH triggered by visual stimulation (Supplementary Fig. 4).

Although we found some defective pixels in our sensor, as shown in the dark pixels in Supplementary Fig. 1b, the probability was 2.89%, which is smaller than that of our previous sensor (5.0%)[7]. Thus, we also succeeded in the reduction of defective pixels.

We found such changes in pH at a micrometer-scale resolution, suggesting that pH changes may be involved in fine-tuning brain activity. This may be clinically important given that the pH in the brains of patients with psychological disorders, such as schizophrenia and bipolar disorder, is abnormal[21]. However, investigating the importance of pH regulation in such situations would require further modification of our sensor to enable chronic implantation and long-term observation. We should also highlight that pH dysfunction is not limited to neurological diseases, since for example, the pH of the environment induced by cancer cells differs greatly to normal healthy cells[22].

In conclusion, we successfully applied a CMOS-based proton image sensor for the observation of biological dynamics of pH,

suggesting the potential application of this sensor in a wide range of biological investigations. Therefore, our proton image sensor may have broad utility in studies that seek to uncover the relationship between cellular pH dysfunction and various pathologies.

## Methods

**Sensor and software development.** The sensor chip was based on our previously described $128 \times 128$ pixel CMOS-based proton image sensor[7], but redesigned to have a slimmer chip width and thickness to minimize tissue damage when inserted into the brains of mice (Fig. 1). The slimmer chip width was achieved by reducing the number of pixels from a $128 \times 128$ matrix to a $128 \times 32$ matrix (Table 1). The reduction in sensor chip thickness was achieved through back-grinding. As in the previous sensor chip, the proton-sensing area of the sensor chip was composed of a 100-nm-thick $Si_3N_4$ film, while its back and lateral sides were waterproofed with a 1-μm-thick SiNx layer. The sensor chip was packaged with a printed circuit board by wire bonds encapsulated in epoxy as shown in Fig. 1a.

The $H^+$ ion sensitivity of our proton image sensor was evaluated using three different solutions with defined and stable pH levels (pH 4.01, pH 6.86, and pH 9.18) held at a constant voltage via a glass electrode[7]. The Si3N4 film of the sensor chip adsorbs protons and thus proton changes are detected as changes in the surface potential (voltage) at each pixel. The voltage readouts from five central pixels of the proton image sensor in these three solutions were used to calculate voltage–pH standard curves (Supplementary Fig. 1a).

**Insertion of the proton image sensor into the visual cortex.** All animal experiments were approved by the National Institute for Physiological Sciences Animal Care and Use Committee (approval number 18A102), and were in accordance with National Institutes of Health guidelines. Male, 8–10 weeks old, C57BL/6 mice housed under a 12-h light/dark cycle with free access to food and water were used for all experiments. Mice were anesthetized using ketamine (74 mg kg$^{-1}$, administered intraperitoneally (i.p.); Daiichi Sankyo, Inc., Tokyo, Japan) and xylazine (10 mg kg$^{-1}$, i.p.; Bayer AG, Leverkusen, Germany) with topically applied 2% xylocaine jelly (AstraZeneca plc Co., Ltd, Cambridge, UK) for further pain management. The scalp was shaved and sterilized with 70% ethanol before the skin and underlying connective tissue were removed to expose the skull. A custom-made stainless steel head plate was attached to the skull with two types of ceramic glass ionomer dental cement (GC Fuji LUTE BC; GC Corp., Tokyo, Japan; Bistite II; Tokuyama Dental, Tokyo, Japan). Once set, the skull was waterproofed with acryl-based dental adhesive resin cement (Super bond; Sun Medical Co. Ltd, Shiga, Japan).

The following day, mice were secured via the custom-made head plate in a stereotaxic frame (SR-6M-HT, Narishige, Co., Ltd, Tokyo, Japan) and anesthetized using urethane (1.5 g kg$^{-1}$, i.p.; Sigma-Aldrich Inc., Missouri, USA). A 2.0 mm × 3.0 mm rectangular piece of skull, and its underlying dura mater membrane, over the left V1 were removed using a dental drill and surgical needle hook. The stereotaxic coordinates for the center of the rectangle were 3.0 mm posterior and 2.5 mm lateral to the Bregma skull landmark.

In order to minimize tissue damage during the pH sensor insertion process, a 1.0 mm deep incision was first made within the left V1 using a sterile microknife (10055-12, Fine Science Tools, Inc. BC, Canada). The proton image sensor was then slowly and carefully inserted into this incision using a micromanipulator (SM-15R, Narishige Co., Ltd). Each stage of the insertion procedure was started only after bleeding had been controlled. This was achieved by continuously washing the area with HEPES-buffered saline (pH 7.4, 150 mM NaCl, 3 mM KCl, 2 mM CaCl$_2$, 1 mM MgCl$_2$, 10 mM glucose, and 5 mM HEPES), with care taken to avoid the formation of large blood clots.

In order to confirm the insertion depth of the proton-sensing area into the brain, the implanted proton image sensor was perpendicularly illuminated with white light (LG-PS2; OLYMPUS Co., Ltd, Tokyo, Japan). Due to the photoelectric effect, the voltage readouts of all proton-sensing area pixels were not inserted into the brain spike dramatically when illuminated by white light (Supplementary Fig. 2a). To correctly detect the sensor area inserted into the brain, the averaged voltage readout during the pre-exposure period (10 s) was subtracted from the averaged voltage readout during white light illumination (10 s) and the area above the brain showed more drastic voltage change (Supplementary Fig. 2b).

**Visual stimulation.** Visual stimuli were presented to mice using an LCD monitor (8 inch, LCD-8000VH, Century, Co., Ltd, Tokyo, Japan) positioned 13 cm away from the right eye. The visual stimuli comprised drifting gratings at eight different directions, essentially black and white bars sweeping across the screen at an angle of 0°, 45°, 90°, 135°, 180°, 225°, 270°, or 315° (ref. [23]). One visual stimulus trial consisted of a 2 s presentation of one of the drifting gratings followed by an 8 s presentation of a gray screen (Fig. 2). In total, each of the eight drifting gratings was presented as 30 trials in a randomized order. Before the first visual stimulus trial only, mice were presented with a gray screen for 120 s. The timing of the visual stimulus trials was controlled and recorded using an NI DAQ device (National Instruments Co., Ltd) controlled by LabVIEW (version 2014; National Instruments Co., Ltd, Texas, USA) and custom-written routines executed in MATLAB (R2017b; MathWorks, Inc., Massachusetts, USA). The pH in the left V1 was continuously measured using the implanted proton image sensor throughout the entire protocol of thus visual experience task.

**Histology.** After the proton imaging experiments, mice were anesthetized using ketamine (74 mg kg$^{-1}$, i.p.) and xylazine (10 mg kg$^{-1}$, i.p.) and terminally perfused with 4% paraformaldehyde. Harvested brains were sagittally sliced at a thickness of 50–100 μm using a Leica VT1000S vibratome (Leica microbiosystems, Wetzlar, Germany). The correct placement of the proton image sensor in the V1 was confirmed by viewing the brain slices using a Nikon A1R microscope (Nikon, Tokyo, Japan).

**Data analysis for pH changes during visual stimulation.** All proton imaging data were analyzed using MATLAB (Mathworks). As the proton image sensor measures pH through its effect on the surface potential at each pixel, their individual voltage readouts were first converted to pH values using voltage–pH standard curves (see section Sensor and software development).

The following calculations were performed at each pixel. Using the mean pH during stimulus interval (i.e. the time between two consecutive visual stimulation trials) at each trial ($pH_{interval, trial}$), the mean pH during the interval at each angle ($pH_{interval, \theta}$) was calculated using Eq. (1):

$$pH_{interval,\theta} = \sum_{trial=1}^{30} \frac{pH_{interval, trial}}{30}. \qquad (1)$$

Using the obtained $pH_{interval, \theta}$ in Eq. (1) the mean pH during the interval over all trials and angles ($pH_{interval}$) was calculated using Eq. (2):

$$pH_{interval} = \frac{pH_{interval, 0°} + pH_{interval, 45°+...} + pH_{interval, 315°}}{8}. \qquad (2)$$

Using the obtained $pH_{interval}$ in Eq. (2), delta pH by visual stimulation at each trial at each angle ($\Delta pH_{trial}$) was calculated by subtraction of the $pH_{interval}$ from the mean pH during stimulation at each trial at each angle ($pH_{stim, trial}$) using Eq. (3):

$$\Delta pH_{trial} = pH_{stim, trial} - pH_{interval}. \qquad (3)$$

Using the obtained $\Delta pH_{trial}$ in Eq. (3), the mean delta pH at each angle ($\Delta pH_{\theta}$) were calculated using Eq. (4):

$$\Delta pH_{\theta} = \sum_{trial=1}^{30} \frac{\Delta pH_{trial}}{30}. \qquad (4)$$

Using the obtained $\Delta pH_{\theta}$ in Eq. (4), the mean delta pH over all trials and angles (i.e. mean in Fig. 2) was calculated using Eq. (5):

$$\Delta pH = \frac{\Delta pH_{0°} + \Delta pH_{45°+...} + \Delta pH_{315°}}{8}. \qquad (5)$$

Using the obtained $\Delta pH_{\theta}$ in Eq. (4) and $\Delta pH$ in Eq. (5), the mean delta–delta pH at each angle ($\Delta(\Delta pH)$, visual response at each angle in Fig. 2) was calculated using Eq. (6):

$$\Delta(\Delta pH) = \Delta pH_{\theta} - \Delta pH. \qquad (6)$$

To confirm that $\Delta(\Delta pH)$ in Eq. (6) were not signal artifacts but truly reflected actual changes in brain pH, the same visual experience task was also performed with the proton image sensor placed in HEPES-buffered saline instead of in live mice (Figs. 2d, 3b, 4b and 5). The pixel-by-pixel distributions of $\Delta(\Delta pH)$ were plotted as histograms in Fig. 3. The 95% confidence intervals for these HEPES-buffered-saline delta pH values were used as thresholds for determining whether a pH change at a given pixel was a signal artifact or reflected actual brain pH (Fig. 3). In other words, $\Delta(\Delta pH)$ recorded in a mouse's brain was only considered to be an actual change in brain pH if it exceeded these 95% confidence interval values. The percentage of pixels which qualified as a true biological signal using this criterion in both mice and HEPES-buffered saline samples are plotted as box and whisker plots in Fig. 4.

Using the obtained $\Delta pH_{trial}$ in Eq. (3) and $\Delta pH$ in Eq. (5), delta–delta pH at each trial at each angle were calculated ($\Delta(\Delta pH_{trial})$, visual response at each trial at each angle) using Eq. (7):

$$\Delta(\Delta pH_{trial}) = \Delta pH_{trial} - \Delta pH. \qquad (7)$$

To evaluate the change statistically, the 30 obtained values of $\Delta(\Delta pH_{trial})$ in Eq. (7) were compared to $\Delta pH$ in Eq. (5) at each angle using a one-sample $t$-test in each pixel (shown in Fig. 5a).

To investigate the temporal dynamics of the change in pH triggered by the visual stimulation, we summarized the data in peri-stimulus time histograms to visualize the dynamics before and after the onset of the visual stimulation (Supplementary Fig. 4). Only data of the pixels recorded in the brain (for example, below the surface, see Fig. 2) were analyzed. We calculated the mean pH dynamics of each pixel before, during, and after the visual stimulation, over multiple trials (in Supplementary Fig. 4a referred to as pre-stim (2 s interval before each stimulation), stim (2 s stimulation), post-stim (4 s interval after each stimulation), respectively). The response pattern at each pixel was categorized as an alkaline response, acidic response, or neutral response based on whether the pH was statistically increased, decreased, or unchanged by visual stimulation ($p < 0.05$, two-sample $t$-test for pre-stim versus stim across 30 trials for each angle). This clearly indicated the

temporally dynamic change in pH, at each pixel. Data for different pixels were sorted according to the response categories and median values during visual stimulation (Supplementary Fig. 4a). To further quantify these responses in the brain, we summarized all response patterns over all angles and animals for each response category (Supplementary Fig. 4b), and calculated the time constant of acidic and alkaline changes in brain pH during visual stimulation ($\tau_{alkali,\ fast} = 250.2$ ms, $\tau_{alkali,\ slow} = 14.19$ s, $\tau_{acid,\ fast} = 231.0$ ms, $\tau_{acid,\ slow} = 6.99$ s). We used two component exponential curve fitting[24] to calculate response patterns over all mouse samples ($n = 9$, Supplementary Fig. 4b).

**Statistics**. Statistical analysis was another method used to identify whether pH changes in individual pixel positions truly reflected actual changes in brain pH. Here, the mean population pH value for each pixel position was calculated by averaging all of the pH values recorded over the 30 visual stimulation trials, for all eight drifting gratings directions across nine mice or across three HEPES-buffered saline samples. A one-sample $t$-test was then used to compare this population mean with the mean pH value recorded for the 30 visual stimulation trials for each of the eight drifting gratings directions. Pixel positions with statistically significant pH changes were then counted for each of the eight drifting gratings directions both in mice and HEPES-buffered saline samples, and the relative abundances of these pixel positions with statistically significant pH changes were compared using a two-way ANOVA test. Subsequently, these pixel positions were pooled across the eight drifting gratings directions and an unpaired $t$-test was used to compare their abundance between mice and HEPES-buffered saline samples. A one-way ANOVA followed by turkey's test was used to compare each distribution of output voltage of the three pH conditions to each other.

**Reporting summary**. Further information on research design is available in the Nature Research Reporting Summary linked to this article.

## Data availability

Data supporting the findings of this work are available within the paper and its Supplementary Information files. The datasets generated during and/or analyzed during the current study are available from the author upon request. The source data underlying Figs. 2c, d, 4, 5b, c, as well as Supplementary Figs. 1c, d, 3a and 4b are provided as a Source Data file, which is also available at https://figshare.com/s/87f676569c5cc2f6cfe0.

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

## Acknowledgements

This work was supported by JST CREST Grant Number JPMJCR14G2, Japan. We thank Ikuko Takeda (Division of Homeostatic Development, National Institute for Physiological Science) for providing us with the illustrations (Fig. 2a and Supplementary Fig. 2a) and Shigenori Inagaki (Department of Developmental Neurophysiology, Kyushu University) for giving us useful suggestion for calculation of constant value (Supplementary Fig. 4d).

## Author contributions

H.H., M.A., K.S., and J.N. designed research; H.H., J.I., Y.N., and S.N. performed research; H.H. and M.A. analyzed data; H.H., M.A., Y.N., D.L.C., Y.K., T.I., K.T., K.S., and J.N. wrote the paper.

## Competing interests

The authors declare no competing interests.
