## [Peer Review File · Nature Communications]

Reviewers' comments:

Reviewer #1 (Remarks to the Author):

The authors report on a CMOS-based proton camera with 128 x 32 pH sensors with a pitch of 23.55 μm and a full frame rate of 50 Hz for in-vivo applications and show related experimental biological data recorded from mice.

Concerning the CMOS device as such the amount of novelty is limited as the chip used here is a derivative of a former CMOS proton camera from the same authors (IEEE TED 2013, ref. 6) but with a four times higher number of sensor sites (not aiming for in-vivo applications), so that in-vivo application and related results are the most interesting topics of this work.

Unfortunately, some major and minor, in part formal issues are found which should be carefully reworked by the authors before the paper can be considered for publication in the journal.

A more detailed list is given below:

1. Line 69, width of the inserted chip:

To allow in-vivo application, the former proton camera chip is shrunk to a width of 1.76 mm. The discussion on suitable widths and volumes of implantable electronic tools is well known from the various needle devices for AP and LFP recording and in-vivo neural tissue stimulation.

Compared to these and compared to the dimensions of a mouse head, 1.76 mm seems relatively wide.

Can you briefly comment on this in the manuscript?

2. Line 75:

“calculated” could perhaps better read “determined”.

3. Line 78 and Fig. S1b:

The authors claim “low intra-pixel variation”. Given the color code ranging from pH 4 to pH 10 the figure is not suitable for an interpretation what “low” quantitatively means.

Plotting the empirical distribution would probably far better help to depict the variation.

In any case, please provide quantitative values in the manuscript for the standard deviations of the obtained voltages (or directly converted to pH values) for the three pH values used in this consideration.

4. Fig. S1b, dark area at the bottom in all sub-figures:

Could you please briefly comment on that? Systematic error?

5. Fig. 4:

It is confusing, that the 0 % whisker mark or the 25 % lower line of the box are sometimes below 0% of the y-axes of the plots.

Please correct or carefully re-explain if the presentation of that data is correct.

6. Fig. captions of Figs. 4 and 5:

Both captions start with a full sentence revealing a statement and do not describe what the figure shows.

Please re-phrase and use the same style as for the captions of Figs. 1, 2, and 3.

7. Compared to the IEEE TED paper as of 2013, a higher sensitivity is achieved here.

Could you please briefly comment on that, perhaps in the supplementary material?

8. Table:

The expressions in the left column should better read “Number of pixels” (plural), “Size of pixel [μm^2] ...”, “Size of sensor chip [mm^3] ...”, and “Frame rate [frames/sec]”, respectively.

9. English:

In the manuscript, sometimes present, sometimes past tense is used.

Please consider using present tense throughout the entire manuscript.

10. Reference 6:

Please use 4 capitals for "IEEE".

Reviewer #2 (Remarks to the Author):

The manuscript "A novel CMOS-based bio-image sensor to spatially resolve neural activity dependent proton dynamics in the awake brain" describes the development of a novel implantable sensor used to study pH dynamics in mice. This is a well written and interesting manuscript that may provide new insights into the temporal dynamics of pH fluctuations in the brain. There were a few minor points that the authors should address in a revision of the current manuscript. In addition, I was unable to find any supplemental materials for the manuscript, which appear to be referenced. Outlined below are the issues that should be addressed in a revised application.

1. The authors state the temporal resolution of the detector but only focus on the spatial resolution of the sensor. Some data showing the temporal dynamics of the sensor and validation of the stated temporal resolution should be included.

2. An advantage of the CMOS sensor is its spatial resolution but a limitation is the spatial coverage. This should be mentioned in the manuscript since the pH sensitive MRI markers (T1rho and CEST) provide whole brain coverage.

3. It is not clear how many mice were studied. This needs to be stated clearly in the manuscript. It is only mentioned in the figure caption for Fig 5b

4. It is not clear how the Figures showing the response of the pH sensor to the gratings were generated (2c 5a). Were these from a single mouse or the average over all of the mice studied. If this was an average, how were the sensors aligned across mice?

5. It would be nice to see the overlap of the sensor placement within mice.

6. The authors use the word slimmer, but it may be better to use smaller / miniaturized or another synonym.

7. Please check that Supplemental material was submitted along with the manuscript.

8. There were a few grammatical changes needed:

8.1 Lines 37-38: exchanger should be exchange

8.2 Line 68: suitable for in vivo measurements

Point-by-point response to the reviewers

We thank the reviewers for considering our manuscript for publication in *Nature Communications*, and for their feedback. We believe that the revised manuscript substantially addresses all the points raised by the reviewer, and has been improved/corrected.

We summarized our point-by-point responses to the reviewers' suggestions and comments here. We added or modified texts and figures to reflect those points, with some grammatical modification. Modified texts are highlighted by yellow in the current manuscript.

Reviewers' comments:

Reviewer #1 (Remarks to the Author):

The authors report on a CMOS-based proton camera with 128 x 32 pH sensors with a pitch of 23.55 μm and a full frame rate of 50 Hz for in-vivo applications and show related experimental biological data recorded from mice.

Concerning the CMOS device as such the amount of novelty is limited as the chip used here is a derivative of a former CMOS proton camera from the same authors (IEEE TED 2013, ref. 6) but with a four times higher number of sensor sites (not aiming for in-vivo applications), so that in-vivo application and related results are the most interesting topics of this work.

Unfortunately, some major and minor, in part formal issues are found which should be carefully re-worked by the authors before the paper can be considered for publication in the journal.

We thank the reviewer #1 for the comments and suggestions. We carefully addressed the specific issues below.

A more detailed list is given below:

1. Line 69, width of the inserted chip:

To allow in-vivo application, the former proton camera chip is shrunk to a width of 1.76 mm. The discussion on suitable widths and volumes of implantable electronic tools is well known from the various needle devices for AP and LFP recording and in-vivo neural tissue stimulation.

Compared to these and compared to the dimensions of a mouse head, 1.76 mm seems

relatively wide.

Can you briefly comment on this in the manuscript?

We have added discussion for the points below in the discussion part of our manuscript (line 171).

To minimize the tissue damage, we miniaturized the device down to the 0.1 mm thickness, while, to detect the change in pH in a wide brain area, we kept 128×32 pixel sensing area. This resulted in a 1.76 mm width of our device, which is still smaller than the previous 128×128 pixel sensor. Although it is larger in comparison to some of the needle devices, the application of these devices is limited to the single point recording or stimulation. Instruments for the wide-field observation or recording with high spatial resolution tends to be larger (Jennings et al, Nature, 2019; Meng et al, eLIFE, 2019; Wang et al, Journal of neural engineering, 2012; Voigts et al, Frontiers in systems neuroscience, 2013; Nicolelis et al, PNAS, 2013; Brochier et al, Scientific data, 2018). For example, the GRIN lens, which has been broadly used to investigate the neural function in the deep and wide area of the brain (such as the amygdala, hippocampus, etc.), is usually of 0.6-1.0 mm diameter and inserted into the 2.0-5.0 mm or deeper areas (Jennings et al, Nature, 2019; Meng et al, eLIFE, 2019). Our device (0.176 mm^2 in cross-section) is theoretically less invasive than the regular GRIN lens ($0.283\text{-}0.785 \text{ mm}^2$).

2. Line 75:

“calculated” could perhaps better read “determined”.

We thank the reviewers, and we have replaced “calculated” with “determined” in the present manuscript (line 78).

3. Line 78 and Fig. S1b:

The authors claim “low intra-pixel variation”. Given the color code ranging from pH 4 to pH 10 the figure is not suitable for an interpretation what “low” quantitatively means. Plotting the empirical distribution would probably far better help to depict the variation. In any case, please provide quantitative values in the manuscript for the standard deviations of the obtained voltages (or directly converted to pH values) for the three pH values used in this consideration.

We apologize that we wrote “intra-pixel” instead of “inter-pixel” incorrectly. We agree that the figure is not enough to quantitatively prove “low inter-pixel variation”. To further address this point, we newly prepared Fig. S1c and S1d. In Fig. S1c, we showed the distribution of output-voltage over all pixels in a sensor. Groups of pixel by pixel values of the three different pH conditions were not overlapped (and statistically

distinct) each other. We further calculated the standard deviation over multiple devices (n=12 sensors in total; SD = 0.0061 ± 0.0013 (pH 10.01), 0.0037 ± 0.0027 (pH 6.86), 0.0171 ± 0.0117 (pH 4.01)) as illustrated in Fig. S1d). These “inter-pixel variations” were relatively small when compared to the value (SD = 0.027) shown in the previous report (Moser et al., IEEE Transactions on Biomedical Circuits and Systems, 2018). The reduction of defective pixels in our new sensor.

We have explained it in the legend of Fig. S1b and the discussion part of the manuscript (line 80).

5. Fig. 4:

It is confusing, that the 0 % whisker mark or the 25 % lower line of the box are sometimes below 0% of the y-axes of the plots.

Please correct or carefully re-explain if the presentation of that data is correct.

We thank for the comment. In Fig. 4, we previously subtracted values of negative control (i.e. data of HEPES) from the value in the brain and the HEPES. But we realized that this is confusing, as suggested. Thus, we decided to alternatively show raw “% pixels” (i.e. without subtraction) in the revised figure. Please confirm that there are no value below 0% in the present form.

6. Fig. captions of Figs. 4 and 5:

Both captions start with a full sentence revealing a statement and do not describe what the figure shows.

Please re-phrase and use the same style as for the captions of Figs. 1, 2, and 3.

We thank the comment, and modified them as suggested.

7. Compared to the IEEE TED paper as of 2013, a higher sensitivity is achieved here.

Could you please briefly comment on that, perhaps in the supplementary material?

We thank for the advice. As suggested, the sensitivity of our sensor has been improved from 32.8 mV/pH to 51.6 mV/pH compared to previous 128×128 sensor. We mentioned this point in the present supplementary material and the result section of the manuscript (line 80).

8. Table:

The expressions in the left column should better read “Number of pixels” (plural), “Size of pixel [μm^2] ...”, “Size of sensor chip [mm^3] ...”, and “Frame rate [frames/sec]”, respectively.

We thank the suggestion, we have followed the reviewer's suggestion as shown in line 553

9. English:

In the manuscript, sometimes present, sometimes past tense is used.

Please consider using present tense throughout the entire manuscript.

We thank the advice. The manuscript has been carefully revised by a professional language editing service to improve the grammar and readability.

10. Reference 6:

Please use 4 capitals for "IEEE".

We have corrected it.

Reviewer #2 (Remarks to the Author):

The manuscript "A novel CMOS-based bio-image sensor to spatially resolve neural activity dependent proton dynamics in the awake brain" describes the development of a novel implantable sensor used to study pH dynamics in mice. This is a well written and interesting manuscript that may provide new insights into the temporal dynamics of pH fluctuations in the brain. There were a few minor points that the authors should address in a revision of the current manuscript. In addition, I was unable to find any supplemental materials for the manuscript, which appear to be referenced. Outlined below are the issues that should be addressed in a revised application.

We thank the reviewer's constructive comments. We have further addressed to the specific issues below.

1. The authors state the temporal resolution of the detector but only focus on the spatial resolution of the sensor. Some data showing the temporal dynamics of the sensor and validation of the stated temporal resolution should be included.

To validate the temporal resolution of our sensor, we summarized *in vivo* recording data to visualize dynamic change triggered by visual stimulation (Fig. S4a, S4b). Details are explained below.

In Fig. S4a, we showed pH dynamics at each pixel, before/during/after the visual stimulation (averaged over multiple trials), clearly demonstrating the temporally dynamic change in pH caused by the visual stimulation. Pixel-by-pixel variation was also visualized in the figure. To further quantify these responses in the brain, we summarized

all response patterns from all angles and animals for each response category (i.e. acidic or alkaline separately) as shown in Fig. S4b, and calculated the time constant of acidic and alkaline change in brain pH during visual stimulation ($\tau_{\text{alkali, fast}} = 250.2$ msec, $\tau_{\text{alkali, slow}} = 14.19$ sec, $\tau_{\text{acid, fast}} = 231.0$ msec, $\tau_{\text{acid, slow}} = 6.99$ sec). These results demonstrate that our sensor successfully detected the sub-second order temporal change in brain pH during visual stimulation. (This is also explained at line 153, 338)

2. An advantage of the CMOS sensor is its spatial resolution but a limitation is the spatial coverage. This should be mentioned in the manuscript since the pH sensitive MRI markers (T1rho and CEST) provide whole brain coverage.

We discussed this point more carefully in the discussion section of the revised manuscript (line 185).

We agree that the spatial coverage of our sensor is smaller than that of the MRI (MRI: whole brain (220×220 mm = 4.84×10^4 mm²), our sensor: 0.72 mm \times 3.0 mm = 2.16 mm²), while our sensor is advantageous for the spatial and temporal resolution (MRI: ~ 4.0 mm, ~ 0.17 frame/sec, our sensor: 23.55 μ m, 50 frame/sec). Also, the spatial coverage of our sensor is larger than regular 2 photon microscopy (2PMS), an imaging technique for the deep brain recording, or single photon imaging through a GRIN lens (our sensor: 2.16 mm², 2PMS: ~ 0.1 mm², GRIN lens: ~ 0.785 mm² with more invasion). Our sensor is also superior to the other imaging techniques in terms of temporal resolution (our sensor: 50 frame/sec, 2PMS: 30 frame/sec, MRI: ~ 0.17 frame/sec, ³¹P spectrometry: several minutes/frame).

3. It is not clear how many mice were studied. This needs to be stated clearly in the manuscript. It is only mentioned in the figure caption for Fig 5b

We carefully revised our manuscript to explicitly indicate the number of mice (line 134, 164, 353, 360, 524, 544).

4. It is not clear how the Figures showing the response of the pH sensor to the gratings were generated (2c 5a). Were these from a single mouse or the average over all of the mice studied. If this was an average, how were the sensors aligned across mice?

We modified the figure legend (line 492, 533) to address this point. Fig 2c, 5a show representative responses of the same single mouse, which are also more carefully explained in the current manuscript. In contrast, the results of all animals were summarized in Fig. 3, 4, and 5b-c. In addition, following the policy of the Nature Communications, we have added individual data points in modified Fig. 4, 5. We also

modified the texts in the method section to more clearly explain the procedure (line 289).

5. It would be nice to see the overlap of the sensor placement within mice.

We thank for the suggestion. We carefully described how the sensor was implanted within the mouse brain (Fig. 2a and Fig. S2a).

6. The authors use the word slimmer, but it may be better to use smaller / miniaturized or another synonym.

As suggested, we replaced the term with “miniaturized”.

7. Please check that Supplemental material was submitted along with the manuscript.

Since the other reviewer seemed to review our supplemental materials correctly, we assume that our submission was successfully processed. We will further ask the editor to confirm that the reviewer #2 could successfully receive it.

8. There were a few grammatical changes needed:

8.1 Lines 37-38: exchanger should be exchange

We thank this comment, but Na^+/H^+ exchanger, Na^+ driven $\text{Cl}^-/\text{HCO}_3^-$ exchanger and passive $\text{Cl}^-/\text{HCO}_3^-$ exchanger are the name of transporters. Thus, we left it as it is.

8.2 Line 68: suitable for in vivo measurements

We thank this comment. We corrected “in vivo” to “in vivo measurements” in the revised manuscript.